**Data Availability Statement:** All relevant data are within the manuscript and its Supporting Information files. The original database available in

# Pediatric emergency care in a low-income country: Characteristics and outcomes of presentations to a tertiary-care emergency department in Mozambique

**Valentina Brugnolaro**[1]*, **Laura Nai Fovino**[1], **Serena Calgaro**[1], **Giovanni Putoto**[2], **Arlindo Rosario Muhelo**[3], **Dario Gregori**[4], **Danila Azzolina**[4], **Silvia Bressan**[1,5], **Liviana Da Dalt**[1,5]

**1** Pediatric Residency Program, Department of Woman's and Child's Health, University of Padova, Padova, Italy, **2** Doctors with Africa CUAMM, Padova, Italy, **3** Pediatric Department, Beira Central Hospital, Beira, Mozambique, **4** Epidemiology and Public Health, Department of Cardiac, Thoracic and Vascular Sciences and Public Health, University of Padova, Padova, Italy, **5** Pediatric Emergency Unit, Department of Woman's and Child's Health, University of Padova, Padova, Italy

* valentina.brugnolaro@gmail.com

## Abstract

### Background

An effective pediatric emergency care (PEC) system is key to reduce pediatric mortality in low-income countries. While data on pediatric emergencies from these countries can drive the development and adjustment of such a system, they are very scant, especially from Africa. We aimed to describe the characteristics and outcomes of presentations to a tertiary-care Pediatric Emergency Department (PED) in Mozambique.

### Methods

We retrospectively reviewed PED presentations to the "Hospital Central da Beira" between April 2017 and March 2018. Multivariable logistic regression was used to identify predictors of hospitalization and death.

### Results

We retrieved 24,844 presentations. The median age was 3 years (IQR 1-7 years), and 92% lived in the urban area. Complaints were injury-related in 33% of cases and medical in 67%. Data on presenting complaints (retrieved from hospital paper-based registries) were available for 14,204 (57.2%) records. Of these, respiratory diseases (29.3%), fever (26.7%), and gastrointestinal disorders (14.2%) were the most common. Overall, 4,997 (20.1%) encounters resulted in hospitalization. Mortality in the PED was 1.6% (62% ≤4 hours from arrival) and was the highest in neonates (16%; 89% ≤4 hours from arrival). A younger age, especially younger than 28 days, living in the extra-urban area and being referred to the PED by a health care provider were all significantly associated with both hospitalization and death in the PED at the multivariable analysis.

DANS public repository as "dataset 'Access to Beira PED from April 2017 to March 2018'": https://doi.org/10.17026/dans-zf9-xwzp.

**Funding:** The author(s) received no specific funding for this work.

**Competing interests:** The authors have declared that no competing interests exist.

## Conclusions

Injuries were a common presentation to a referral PED in Mozambique. Hospitalization rate and mortality in the PED were high, with neonates being the most vulnerable. Optimization of data registration will be key to obtain more accurate data to learn from and guide the development of PEC in Mozambique. Our data can help build an effective PEC system tailored to the local needs.

## Introduction

Over the last two decades, child mortality significantly decreased worldwide thanks to the development of the Millennium Development Goals (MDGs) and the Sustainable Development Goals (SDGs), elaborated by the United Nations to reduce healthcare disparities [1–5]. However, child mortality remains high in low-income countries (LICs) and, in particular, in Sub-Saharan Africa [6, 7].

The development of efficacious pediatric emergency care services has been identified as one of the crucial steps to reduce child mortality [8–10]. However, pediatric care services still represent the weakest links in the healthcare systems chain [11–13], with pediatric emergency medicine still being an understudied field [14, 15].

The study of the burden and profile of pediatric emergencies is important to understand how to optimize resource allocation and healthcare facilities to develop a structured emergency care system that could further reduce child mortality.

In Mozambique, as in other sub-Saharan countries, the health care system is extremely diverse. Mozambique has approximately 1,600 healthcare facilities (including health posts, health centers, district hospitals, provincial hospitals, and four referral/central hospitals) distributed in 11 provinces, 30 municipalities, and 157 provinces. Overall, 96% of these facilities only deliver primary care (i.e. essential preventive and curative health). Only two of the four tertiary-care referral hospitals have a Pediatric Emergency Department (PED), one of which is in Beira, the capital of the Province of Sofala. In 2017, the under-5 and neonatal mortality rates in Sofala were 75.6 per 1,000 and 25 per 1,000 live births, respectively [16–19].

As for other LICs, data on the epidemiology of pediatric presentations to the PED is important to develop tailored strategies to improve the management of acutely and critically ill children, in order to further reduce child mortality.

The present study aims to describe the profile and outcomes of pediatric presentations to a referral care PED in Mozambique over one year. We also aimed to identify predictors of hospitalization and death, eventually suggesting strategies to improve pediatric emergency care services.

## Materials and methods

### Study design and population

We retrospectively collected data on presentations of all children accessing the PED at Hospital Central da Beira (HCB, Beira, Mozambique), over a 12-month period, between April 2017 and March 2018. The upper age limit for patients' inclusion was 15 years. The study was approved by the ethical review committee of Hospital Central da Beira.

## Study setting: Healthcare in Beira and the Sofala province

Beira is the second largest city of Mozambique (with approximately 530.000 inhabitants over an area of 633 km$^2$) and the capital of the Sofala province (with approximately 2.3 million inhabitants, over an area of 68.018 km$^2$), which includes 159 health centers and posts (13 in the urban area of Beira), four rural/district hospitals, and one referral Hospital (HCB). By merely averaging the overall distribution of healthcare facilities in the Sofala province, we obtain a health facility every 450 km$^2$ in the extra-urban area and one every 45km$^2$ in the urban area. At the time of the study an ambulance service was not available, and patients could reach health care facilities on foot, with private transport or with public transport, where available.

The HCB hosts a Pediatric Department with approximately 200 beds, and one of the two PEDs of the country. The wards are mostly staffed with generalists and pediatric residents, while only seven Pediatricians run the whole Pediatric Department. In the PED some beds are available for short-stay observation and a room is dedicated to Pediatric Intensive Care, where critical children are admitted once stabilized. The Pediatric Department also provides neonatal care to approximately 6000 newborns/year. The Neonatal Intensive Care Unit (NICU) counts about 30 beds and 2000 hospitalizations/year. Based on the most recent available data from 2017, in-hospital overall pediatric mortality was 13%, while in the NICU mortality was 33%.

## Sources of data and data collection procedures

Demographic, clinical, and outcome data of all presentations to the PED were abstracted from three hospital paper registries: i) the "presentations registry"; ii) the "hospitalizations registry"; and iii) the "deaths registry". Death and hospitalization registries were filled by PED medical personnel, while the presentation registry was filled by hospital administrative personnel. Details on data systematically recorded in each of the registries are reported in Fig 1. We were unable to collect data on interventions, treatments, or resuscitations in the PED because this information was reported on paper charts, which were not filed systematically and were therefore unavailable to either clinical or research staff. Also, reliable information on comorbidities was not available from the registries.

All the registries were reviewed by two of the Authors (VB and LNF) and patient identity was crossed-matched between registries. Data were entered into an electronic standard data collection system. Abstractors were trained locally based on the initial review of 200 registry records each. A two-month data abstraction overlap between the abstractors helped in ensuring consistency in data abstraction and coding, by training of the second data abstractor. No formal double entry of data by the two abstractors occurred during this time.

The following data were collected from the registries: sex, age, area of residence, modality of presentation, presenting complaint, outcome (discharge, hospitalization, or death, and time

| Type of Registry | Presentations registry<br>5 paper registries | Hospitalizations registry *<br>3 paper registries | Deaths registry<br>1 paper registry |
|---|---|---|---|
| Type of data systematically recorded in the registries | Name of patient<br>Date of visit<br>Age<br>Sex<br>Area of residence<br>Modality of presentation<br>Presenting complaint<br>Date of discharge | Name of patient<br>Date of hospitalization<br>Diagnosis of hospitalization<br>Ward of hospitalization | Name of patient<br>Time of death (<4h or >4 h from arrival)<br>Date of death |
| Number of total records | 24,884 | 4,997 | 396 |

**Fig 1. Characteristics of hospital registries from which study data were collected.**

of death). Children's age ranged from 0 to 15 years, and the age variable was categorized in four age groups for analysis: from 0 to 28 days (neonates), from 29 days to 1 year (infants), from 1 to 5 years (preschoolers), and from 5 to 15 years (school-aged). The modality of presentation included self-presentations or referrals from other health care providers (i.e. health care centers, rural hospitals, private clinics, etc.). The area of residence variable was categorized in urban, when the child lived in the Beira urban area, and in extra-urban, which was sub-categorized into within the Sofala province and outside the Sofala province. Information on presenting complaints was categorized in the registries as medical or injury related. Medical complaints included the following locally predefined categories of non-traumatic complaints: fever of any origin, respiratory, gastrointestinal, cardiovascular, neurological, musculoskeletal, constitutional, sense organs (which included medical complaints to the eyes, ears, nose, throat and to the skin), and others (which included psychiatric disorders, genitourinary disorders, etc.). Injury-related presentations were locally classified into road accidents, falls, wounds, violence, inhalation/ingestion, burn, and drowning. This categorization system was maintained for data analyses in the current study. Time from presentation to death was categorized in early death (death on arrival or within four hours from arrival), and later death in the PED (after four hours from arrival). Unfortunately, the HBC did not have the facilities and resources (i.e., trained staff, information technology infrastructure) to code diagnoses according to the ICD 9/10 codes. Data on diagnosis were reported as per local documentation practices.

## Statistical analysis

Descriptive statistics were reported as median and interquartile range (IQR) for continuous variables. Categorical variables were reported as proportions and percentages. The Wilcoxon test was used for comparison of continuous variables, while Chi-square and Fisher's exact tests, as appropriate, were used for categorical variables.

We then fit univariable logistic regression models specifying hospitalization, death, and death within four hours as the dependent variable, and clinical and demographic variables as independent variables. Subsequently, we fit multivariable logistic regression models to identify independent predictors of hospitalization, mortality, and mortality within four hours, with the variables that were found to be significant from the univariable models (p<0,05). Results of dependent variables analysis were tested again in a multivariable model for interaction with age, sex, area of residence, the modality of presentation (self vs referred presentation), the reason for presentation, and outcomes [20].

Given the high rate of missing data for the independent variable "presenting complaint" a sensitivity analysis was carried out to assess how missing values would affect the association of the independent variables with the dependent variable. With this respect we performed the following analyses:

1. A complete case analysis estimating the model on the valid case data (excluding records with missing data for the variable presenting complaint)

2. A missing data imputation analysis based on the model estimation on an imputed dataset. A Multiple Imputation by Chained Equations (MICE) procedure was used to handle the missing data for the variable presenting complaint.

3. An estimation of the multivariable model for the outcomes, but excluding the variable presenting complaint from the model.

When the p-value was <.05, the difference was regarded as statistically significant. All statistical tests were 2-tailed. All statistical analyses were performed using Stata Version 13.0

(StataCorp, College Station, TX) and R 3.6.2 together with caret, rms, and MICE packages [21–24].

## Results

### Patients characteristics

During the 12-month study period, 24,844 presentations were recorded. Of these, 14,448 (58.8%) were male, with a male to female ratio of 1.43 to 1. The median age at presentation was 36.5 months (IQR 12–85.2 months). The majority of patients (92%) came from the urban area, with 42% already been assessed by a health care provider in a health care center or at a countryside hospital. A summary of demographics and general characteristics of study presentations by age-group is presented in Table 1.

Data on presenting complaints specifications were available for 14,204 (57.2%) presentations. A medical issue was the reason for presentation in 67% of cases, while 33% were consequent to an injury. The most common medical presentations were respiratory diseases (29.3%), followed by fever (26.7%), and gastrointestinal disorders (14.2%). Among injury presentations, falls (63%) were the most common, followed by foreign body ingestion/inhalation (10.2%) and road accidents (9.8%). Data on presenting complaints by age group are reported in Table 2.

### Outcomes

Overall, 4,997 (20.1%) of encounters resulted in hospitalizations and 396 (1.6%) in death in the PED. Data on outcomes by age group are described in Table 3. Data on length of stay in the PED for patients who were discharged were available for only 5,639 out of 19,451 visits (29.0%). Of these, 5,505 (97.6%) were discharged within 24 hours of arrival.

### Hospitalization analysis

Of the 4,997 hospitalizations, 37 (0.7%) were direct admission from the PED to the PICU. Of these, 17 (45.9%) were for burns. Overall, the length of stay in the PED for visits that resulted in hospitalization was < 24 hours in 37%, between 24 and 48 hours in 56%, between 48 and 72 hours in 5% and > 72 hours in 2% of cases. Data on presenting complaints were available for 4,057 (81.2%) of visits resulting in hospitalization. Of these, 88.1% presented with a medical

**Table 1. Demographic characteristics of Pediatric Emergency Department presentations by age group.**

| Number of visits to the PED | Total | < 28 d | 29 d–1 y | 1–5 y | 5–15 y |
|---|---|---|---|---|---|
| | (n= 24,844) | 2.7% (n = 677) | 22.7% (n = 5,634) | 43.7% (n =10,845) | 30.9% (n = 7,688) |
| Sex * | | | | | |
| Male | **58.8%** | 50.6% (260) | 59.3% (3,315) | 58.3% (6,293) | 59.9% (4,580) |
| Residency * | | | | | |
| Urban (Beira) | **92.5%** | 87.0% (585) | 92.0% (5,189) | 94.0% (10,171) | 91% (7,003) |
| Sofala Province | **6.2%** | 12.0% (82) | 6.0% (338) | 5.0% (569) | 7% (567) |
| Extra Sofala | **1.3%** | 1.0% (9) | 2.0% (105) | 1.0% (102) | 1.0% (115) |
| Modality of Access to the PED | | | | | |
| Self-presentations | **58.0%** | 54.0% (367) | 62.0% (3,466) | 59.0% (6,439) | 45.0% (4,205) |
| Heath Care Center | **36.0%** | 37.0% (249) | 32.0% (1,827) | 36.0% (3,885) | 39.0% (2,976) |
| Peripheral Hospital | **6.0%** | 9.0% (61) | 6.0% (341) | 5.0% (521) | 7.0% (507) |

*Data on sex available for 24,562/24,844 (98.9%); data on residency available for 24,835/24,844 (99.9%).

Data are reported in terms of percentages and absolute frequencies.

Table 2. Presenting complaints by age group.

| PRESENTING COMPLAINT | Total | < 28 d | 29 d–1 y | 1–5 y | 5–15 y |
|---|---|---|---|---|---|
| | n = 14,204 | n= 297 | n = 3,067 | n = 6,232 | n = 4,608 |
| **Injury** | **33.0% (4,682)** | **14.5% (43)** | **16.3% (500)** | **31.1% (1,941)** | **47.7% (2,198)** |
| Fall | 63.0% (2,949) | 39.5% (17) | 61.4% (306) | 60.6% (1,178) | 66.0% (1,448) |
| Ingestion/Inhalation | 10.2% (480) | 14.0% (6) | 10.1% (52) | 16.2% (314) | 5.0% (108) |
| Road Accident | 9.8% (457) | 27.9% (12) | 6.9% (34) | 7.4% (144) | 12.0% (267) |
| Wound | 8.6% (401) | 14.0% (6) | 7.4% (37) | 7.0% (135) | 10.1% (223) |
| Burns | 5.1% (237) | 4.6% (2) | 10.1% (52) | 6.3% (123) | 2.7% (60) |
| Violence | 3.0% (143) | 0.0% (0) | 3.0% (15) | 2.1% (41) | 4.0% (87) |
| Drowning | 0.3% (15) | 0.0% (0) | 0.8% (4) | 0.3% (6) | 0.2% (5) |
| **Medical** | **67.0% (9,522)** | **85.5% (254)** | **83.7% (2,569)** | **68.9% (4,289)** | **52.3% (2,410)** |
| Respiratory | 29.3% (2,789) | 27.2% (69) | 35.6% (914) | 26.3% (1,124) | 28.3% (682) |
| Fever | 26.7% (2,540) | 22.0% (56) | 20.9% (537) | 31.0% (1,328) | 25.7% (619) |
| Gastrointestinal | 14.2% (1,355) | 8.3% (21) | 17.1% (440) | 13.8% (593) | 12.5% (301) |
| Sense Organs* | 9.8% (936) | 13.8% (35) | 9.0% (231) | 9.5% (410) | 10.8% (260) |
| Constitutional** | 8.4% (794) | 20.1% (51) | 8.3% (213) | 8.5% (367) | 6.8% (163) |
| Neurological | 7.9% (752) | 2.7% (7) | 6.7% (171) | 8.3% (355) | 9.1% (219) |
| Musculoskeletal | 2.4% (225) | 3.9% (10) | 1.5% (39) | 1.7% (72) | 4.3% (104) |
| Cardiovascular | 0.8% (77) | 1.6% (4) | 0.6% (15) | 0.4% (16) | 1.7% (42) |
| Other*** | 0.6% (54) | 0.0% (1) | 0.4% (9) | .6% (24) | 0.8% (20) |

* sense organs are defined as the body organs by which humans are able to see, smell, hear, taste and touch or feel. This category includes medical complaints to the eyes, ears, nose, throat and to the skin.

**lethargy, weakness, loss of appetite, fatigue etc.

Data are reported in terms of percentages and absolute frequencies.

complaint, and 11.9% with an injury. A significantly higher proportion of medical presentations were hospitalized compared to injuries (37.9% vs. 9.6%, p-value < 0.001). Results of the univariable analysis assessing the association of available clinical variables with hospitalization is reported in S1 Table.

The multivariable analysis carried out on the subgroup of encounters with data on presenting complaint available and the multivariable analysis with missing data imputation showed similar high odds of being hospitalized if presentations to the PED were due to a medical problem rather than an injury (OR 12.19, 95% CI: 10.78 – 13.38 and 11.79, 95% CI: 10.62-13.1, respectively). (Table 4). A younger age, especially younger than 28 days, living in the extra-urban area and being referred to the PED by a health care provider were all predictors of hospitalization, with similar ORs at all the multivariable analyses performed.

Table 3. Outcomes of pediatric emergency department presentations by age group.

| OUTCOMES | Total | < 28 d | 29 d- 1 y | 1–5 y | 5–15 y |
|---|---|---|---|---|---|
| | n = 24,844 | n = 677 | n = 5,634 | n = 10,845 | n = 7,688 |
| **Mortality** | **1.6% (396)** | **16.1% (109)** | **1.7% (96)** | **1.1% (118)** | **1% (73)** |
| ≤ 4 h | 1.0% (247) | 14.3% (97) | 1% (57) | 0.5% (57) | 0.5% (36) |
| > 4 h | 0.6% (149) | 1.8% (12) | 0.7% (39) | 0.6% (61) | 0.5% (37) |
| **Hospitalization** | 20.1% (4,997) | 29.7% (201) | 25.2% (1,422) | 19,7% (2,132) | 16.1% (1,242) |
| **Discharge** | 78.3% (19,451) | 54.2% (367) | 73.1% (4,116) | 79,2% (8,595) | 82.9% (6,373) |

Data are reported in terms of percentages and absolute frequencies.

**Table 4. Determinants of hospitalization.**

| | | Total of Data Available | Hospitalization | Multivariable analysis on valid cases only* | p-value | Multivariable analysis imputing missing data for the variable presenting complaint | p-value | Multivariable analysis excluding the variable presenting complaint | p – value |
|---|---|---|---|---|---|---|---|---|---|
| | | N | n: 4997 | OR (95% CI) | | OR (95% CI) | | OR (95% CI) | |
| **SEX** | Male | 14,448 | 2,893 | 0.97 (0.89 – 1.05) | 0.43 | 0.99 (0.92 – 1.06) | 0.75 | 0.92 (0.92 – 1.06) | 0.02 |
| | Female | 10,114 | 2,054 | Reference | | Reference | | Reference | |
| **AGE** | | 24,844 | 4,997 | | | | | | |
| | 0 – 28 days | 677 | 201 | 1.81 (1.34-2.43) | <0.001 | 1.68 (1.36-2.07) | <0.001 | 2.8 (2.28-3.43) | <0.001 |
| | 29 d – 1 year | 5,634 | 1,422 | 1.53 (1.36-1.72) | <0.001 | 1.19 (1.08-1.31) | <0.001 | 2.07 (1.89-2.27) | <0.001 |
| | 1 – 5 years | 10,845 | 2,132 | 1.12 (1.01-1.24) | 0.03 | 1.05 (0.96-1.15) | 0.27 | 1.42 (1.31-1.54) | <0.001 |
| | 5 – 15 years | 7,688 | 1,242 | Reference | | Reference | | Reference | |
| **RESIDENCY** | Extra-urban | 1,887 | 818 | 2.97 (2.55-3.45) | < 0.001 | 2.61 (2.31-2.95) | < 0.001 | 2.617 (2.31-2.95) | < 0.001 |
| | Urban | 22,948 | 4,177 | Reference | | Reference | | Reference | |
| **MODALITY OF PRESENTATION** | Health Care Provider referral | 10,360 | 3,380 | 5.57 (5.08-6.11) | < 0.001 | 7.16 (6.63-7.73) | < 0.001 | 3.65 (3.4-3.91) | < 0.001 |
| | Self-Presentations | 14,477 | 1,617 | Reference | | Reference | | Reference | |
| **PRESENTING COMPLAINT** | Medical | 9,522 | 3,607 | 12.19 (10.78-13.78) | < 0.001 | 11.79 (10.62-13.1) | < 0.001 | | |
| | Injury | 4,682 | 450 | Reference | | Reference | | | |

* Missing values for the variable presenting complaint were excluded from the analysis

The valid cases (total of data available) have been reported with the number of hospitalized patients. A sensitivity analysis has been performed reporting the results (OR, 95% Confidence Intervals (CI), and p-values) for 1) Multivariable Analysis on valid cases; 2) Multivariable Analysis on imputed data for variable presenting complaint; 3) Multivariable analysis excluding the variable presenting complaint.

S2 Table describes the univariable association between type of presenting complaint and hospitalization. The multivariable analysis (Table 5) showed that children presenting for a medical complaint had higher odds of being hospitalized if they presented for cardiovascular, constitutional, and neurological complaints compared to fever. Within the injury-related presentations children presenting for burns, road accidents, wounds or ingestion/inhalations had higher odds of being hospitalized compared to children presenting for falls.

The diagnosis distribution for hospitalized patients is described, stratified by age, in Fig 2.

**Mortality analysis.** Overall, mortality in the PED was 1.6%. The majority of deaths (81%) occurred in patients younger than five years, with the highest mortality found in the 0 – 28 days group (16.1%).

Results of the univariable analysis is reported in S1 Table, while those of the multivariable analyses are reported in Table 6. A younger age, especially younger than 28 days, living in the extra-urban area and being referred to the PED by a health care provider were predictors of mortality in the PED at all the multivariable analyses performed, although some variability in ORs for age, modality of presentation and presenting complaint was noticed between the models.

Of the 396 deaths, 247 (62%) occurred within four hours from arrival (early deaths). Due to the low number of patients and the high rate of missing values for the independent variable "presenting complaint" we only performed a multivariable analysis excluding this variable from the model (Table 7).

**Table 5. Association between presenting complaints and hospitalization.**

|  |  | N^ | Hospitalization (n)^^ | Multivariable Analysis OR (95% CI) | p – value |
|---|---|---|---|---|---|
| PRESENTING REASON |  |  |  |  |  |
|  | Medical | 9,522 | 3,607 |  |  |
|  | Fever ‡ | 2,540 | 754 (21%) |  |  |
|  | Respiratory | 2,789 | 873 (24%) | 1.17 (1.03-1.33) | **0.019** |
|  | Neurological | 752 | 465 (13%) | 3.28 (2.72–3.96) | <**0.001** |
|  | Gastrointestinal | 1,355 | 461 (13%) | 1.32 (1.13-1.65) | **0.005** |
|  | Cardiovascular | 77 | 64 (2%) | 7.86 (4.08-15.13) | <**0.001** |
|  | Musculoskeletal | 225 | 81 (2%) | 0.54 (0.39-0.73) | **0.001** |
|  | Constitutional* | 794 | 638 (18%) | 6.98 (5.65 -8.62) | <**0.001** |
|  | Sense Organs** | 936 | 242 (7%) | 0.72 (0.60-0.87) | <**0.001** |
|  | Others*** | 54 | 29 (1%) | 1.99 (1.08-3.69) | **0.028** |
|  | Injury | 4,682 | 450 |  |  |
|  | Drowning | 15 | 1 (0%) | 1.98 (0.25-15.70) | 0.520 |
|  | Road Accident | 457 | 84 (19%) | 4.37 (3.26-5.87) | <**0.001** |
|  | Fall ‡ | 2,949 | 175 (39%) |  |  |
|  | Burn | 237 | 109 (24%) | 25.01(17.99-34.78) | <**0.001** |
|  | Wound | 401 | 36 (8%) | 2.35 (1.60-3.47) | <**0.001** |
|  | Violence | 143 | 3 (1%) | 0.42 (0.13-1.35) | 0.150 |
|  | Ingestion/Inhalation | 480 | 42 (9%) | 2.24 (1.56-3.23) | <**0.001** |

*lethargy, weakness, loss of appetite, fatigue, etc.

** sense organs are defined as the body organs by which humans can see, smell, hear, taste and touch or feel. This category includes medical complaints to the eyes, ears, nose, throat, and the skin.

***Psychiatric and genitourinary diseases.

‡ Reference Category.

Absolute number and percentages of hospitalized patients have been reported. Multivariable and Logistic Regression Model results (OR, 95% Confidence Intervals (CI), and p-values, adjusted for gender, age, residency and modality of presentation) are represented in the table.

A younger age, especially younger than 28 days, was a predictor of early mortality in the PED, while visits that were referred to the PED by a health care provider had lower odds of dying in the first four hours from arrival. Of patients who died after 4 hours, 34% died in the first 24 hours, 43% between 24 and 48 hours, 13% between 48 and 72 hours and 10% beyond 72 hours since arrival.

## Limitations

The results of our study should be interpreted in light of its limitations, which are mostly related to its retrospective design and the available sources of data collection. First, approximately 40% of data on presenting complaints were missing in the "presentation registry". This registry was filled by hospital administrative staff who were less aware of the importance of accurate data completion with respect to reporting and analysis purposes. The rate of missing information on presenting complaints was higher for patients who died in the PED (presenting complaint was not reported in 81.3% of deaths), followed by visits resulting in discharge to home (48.2% of missing information) and those resulting in hospitalization (missing information for 18.8%). While this lack of information affects the accuracy of our findings with respect to the description of presenting complaints, this was the best available data we could get access to at the time of the study. In addition, we performed a sensitivity analysis to report how missing values could have affected our results based on different scenarios. Missing information on

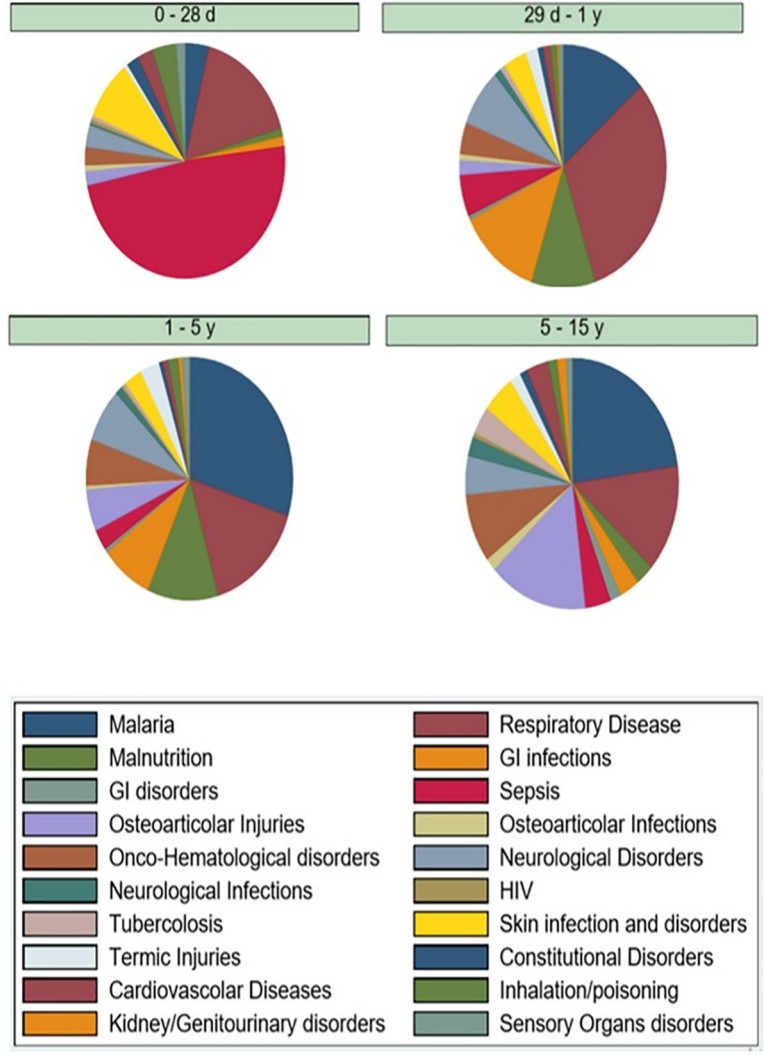

**Fig 2. Hospitalization diagnosis, by age and frequency.**

ED visits records is a common challenge for many LICs and is inherently related to the limitations of the local data registration and repository system. Lack of human and information technology resources represent the major obstacles to the establishment of an accurate and long-lasting data recording and monitoring system in these Countries. Learning from local data is a valuable opportunity for a growing health system to improve the quality of care while optimizing resource use. Efforts towards establishing a robust data management and monitoring system should be made at an institutional and governmental level to best support the development of a pediatric emergency care system in Mozambique.

Second, data on comorbidities were reported inconsistently and only in the hospitalization registry and could not be analyzed. This would be extremely valuable information to include in the multivariable analysis, as underlying conditions such as malnutrition, HIV, and tuberculosis have shown to be associated with the need for hospitalization and mortality [11, 25–27]. Systematic collection of the main comorbidities should be pursued in order to be able to appropriately interpret data to improve care and optimize resource organization and use.

**Table 6.  Determinants of mortality in the Pediatric Emergency Department.**

| | | Total of Data available | Mortality in the PED | Multivariable Analysis on valid cases only* | p-value | Multivariable Analysis imputing missing data for variable presenting complaint | p-value | Multivariable analysis excluding variable presenting complaint | p – value |
|---|---|---|---|---|---|---|---|---|---|
| | | N | (n 396) | OR (95% CI) | | OR (95% CI) | | OR (95% CI) | |
| **SEX** | Male | 14,448 | 182 | 1.02 (0.63–1.65) | 0.94 | 0.96 (0.78 – 1.18) | 0.72 | 0.99 (0.79 – 1.25) | 0.92 |
| | Female | 10,114 | 125 | Reference | | Reference | | Reference | |
| **AGE** | | 24,844 | 396 | | | | | | |
| | 0 – 28 days | 677 | 109 | 5.34 (1.09–15.01) | <**0.001** | 16.12 (11.69-22.22) | <**0.001** | 5.91 (3.06–9.68) | <**0.001** |
| | 29 d – 1 year | 5,634 | 96 | 2.33 (1.25–4.34) | **0.01** | 1.34 (0.99-1.83) | 0.06 | 2.25 (1.65 -3.07) | <**0.001** |
| | 1 – 5 years | 10,845 | 118 | 0.98 (0.54–1.79) | 0.95 | 0.95 (0.71 – 1.28) | 0.74 | 1.31 (0.97 – 1.75) | 0.08 |
| | 5 – 15 years | 7,688 | 73 | Reference | | Reference | | Reference | |
| **RESIDENCY** | Extra-urban | 1,887 | 43 | 2.40 (1.28-4.50) | **0.01** | 2.14 (1.5–3.05) | <**0.001** | 2.11 (1.49–2.98) | <**0.001** |
| | Urban | 22,948 | 353 | Reference | | Reference | | Reference | |
| **MODALITY of PRESENTATION** | Health Care Provider referral | 10,360 | 165 | 7.18 (3.97-12.98) | <**0.001** | 2.36 (1.89 –2.95) | <**0.001** | 1.95 (1.53 –2.47) | <**0.001** |
| | Self-presentation | 14,477 | 231 | Reference | | Reference | | Reference | |
| **PRESENTING COMPLAINT** | Medical | 9,522 | 48 | 2.68 (1.57-4.59) | <**0.001** | 10.36 (6.94-10.45) | <**0.001** | | |
| | Injury | 4,682 | 26 | Reference | | Reference | | | |

*Missing values for the variable presenting complaint were excluded from the analysis.

The valid cases (total of data available) have been reported with the number of deceased patients. A sensitivity analysis has been performed reporting the results (OR, 95% Confidence Intervals (CI), and p-values) for 1) Multivariable Analysis on valid case; 2) Multivariable Analysis on imputed data for variable presenting complaint; 3) Multivariable analysis excluding the variable presenting complaint.

**Table 7.  Determinants for early mortality in the PED ($\leq$ 4h vs mortality> 4h).**

| | | Total of Data available N 28,844 | Early death ($\leq$ 4 h) n: 247 | Multivariable Analysis OR† (95% CI) | p – value |
|---|---|---|---|---|---|
| SEX | **Male** Vs | 14,448 | 90 | 0.91 (0.66 – 1.26) | 0.58 |
| | Female | 10,114 | 71 | | |
| AGE | | 28,844 | 247 | | |
| | 0 – 28 days | 677 | 97 | 2.71 (1.4 – 5.25) | < **0.001** |
| | 29 d – 1 year | 5,634 | 57 | 1.36 (0.88 – 2.09) | 0.160 |
| | 1 – 5 years | 10,845 | 57 | 0.93 (0.61 – 1.43) | 0.753 |
| | 5 – 15 years ‡ | 7,688 | 36 | | |
| RESIDENCY | Urban Vs | 22,948 | 230 | | |
| | **Extra sofala** | 1,887 | 17 | 0.85 (0.50 – 1.44) | 0.727 |
| MODALITY of PRESENTATION | Self-presentations Vs | 14,477 | 178 | | |
| | **Health Care Provider referral** | 10,360 | 69 | 0.35 (0.25 – 0.49) | < **0.001** |

‡Reference category.

The number of valid cases (total of available data) and early death patients has been reported. Logistic Regression Model results (OR, 95% Confidence Intervals (CI), and p-values) are represented in the table.

Third, we could not get access to data prior to April 2017 to identify possible biases in our results from random yearly variations. Based on the local clinical registry filing system, completed paper registries were temporarily filed and available for some months and then periodically burnt.

Fourth, this is a single center study and may not represent the rest of Mozambique or other LICs with different disease prevalence. However, there are only four tertiary care level hospitals in Mozambique, two of which (in Beira and Maputo) have a PED, and it is reasonable to believe our data may provide some useful insights into pediatric emergency care at a local and national level to help optimize distribution and use of resources, as well as plan the most appropriate feasible and effective interventions to improve pediatric emergency care within an integrated system of care.

## Discussion

In this study we were able to provide the first, albeit limited, data on pediatric emergency visits to a tertiary care PED in the LIC of Mozambique. Our data represent a first important step to help the establishment of a pediatric emergency care monitoring system in Mozambique in order to guide the formulation of appropriate strategies to improve the management of the acutely and critically ill children and develop a structured and sustainable emergency care system. The first important finding of our study is that the current PED data registration system has many flaws and challenges, which hamper the provision of accurate and valid data to learn from and guide the development of pediatric emergency care tailored to local needs. Optimization of data registration is an important area on which to focus resources in order to obtain more accurate data for this purpose.

Our study found an overall high mortality rate in the PED setting of 1.6%. Although higher than reported in high-income countries (1.5/100 000 visits) [28], our result is in line or even lower compared to other sub-Saharan countries [29]. The majority of deaths occurred in patients younger than five years (81%), with the highest mortality found in the neonatal group (16%). Based on a recent systematic review [30], about a third of all neonatal deaths tend to occur on the day of birth, and approximately 75% die in the first week of life. These findings suggest that focusing on perinatal care, maternal education and improved access to healthcare is essential for saving newborn lives.

Our analysis also showed that living in extra-urban areas is a predictor of death. This may reflect the many challenges in transportation to the hospital that these children have to face even when severely ill. Fernandes and colleagues [18], highlighted the importance of health service availability, showing an overall improvement in child survival in Mozambique, associated with increased health workforce density, institutional birth coverage, and government health financing, despite the substantial disparity between provinces.

Physicians working at HCB's PED noticed that death occurred more frequently in children presenting late in their course of illness. As evidenced by Punchak and coworkers [31] there are many potential contributing factors to late presentations, including delays related to triage organization, bad tiered health care system, late care-seeking by families due to a lack of health education, and socioeconomic factors related to the geographic distribution of health centers and inadequate transportation infrastructures. Improving access to care, and further promoting health education, would likely result in an earlier presentation to the PED, eventually translating in better disease recognition and treatment.

The majority of deaths in our study occurred in the first four hours from arrival (62%). This group mostly included neonates and children living in the urban area who were brought in by parents. Our study also found that children who had already been evaluated by a health

care provider (in a health care center or a peripheral hospital) had lower odds of dying early (within 4 hours) in the PED. In fact, these children died more often after four hours from arrival to the PED. Our findings may reflect the ability of health care centers or peripheral hospitals to stabilize severely ill patients for transport. However, late presentation to a health care facility and inability to provide effective care during transport may contribute to the unfavorable outcome of these children. Although WHO and UNICEF [6, 7] report that infectious diseases remain a leading cause of death for children under the age of 5 in sub-Saharan Africa, accurate local data would be paramount to better understand which interventions could be most effective to further reduce child mortality both in the PED and at a community level, within an integrated system of care.

We also found a high hospitalization rate of 20.1%, with neonates and infants showing higher odds of being hospitalized compared to older children, as previously described for other sub-Saharan regions [29]. Also, children who lived in the extra-urban area were at higher risk of being hospitalized compared to children who lived in the urban area, especially if already evaluated by health care center physicians or in a peripheral hospital. These results reflect a good organization of the health care referral system in treating pediatric critically ill presentations [11]. However, peripheral health centers have limited resources or lack of training for the management of the most severe presentations, and transfer conditions to the referral center remain challenging.

Our study showed that children presenting for a medical reason were more likely hospitalized, especially if admitted with cardiovascular, constitutional, and neurological diseases. Among injury presentations, burns, road accidents, wounds, and ingestion/inhalations were significantly associated with hospitalization, compared to fall. Based on our field experience, only severe burns with an intrinsic high risk of complications were referred to HBC, which justified the need for hospitalization [31–33]. The causes and risk factors behind the substantial number of severe burns should be further explored in order to implement effective preventive measures.

The analysis of hospitalization diagnosis by age group showed that sepsis, followed by lower respiratory infections (i.e pneumonia and bronchiolitis) and skin infections (i.e cellulitis, impetigo, piodermitis, etc.) were the most common in the neonatal age group. This highlights the need for interventions to improve perinatal care and parents' education. At HBC neonates and their mothers are usually discharged on the first day after delivery, without provision of any further assistance from health personnel in the out of hospital setting. Indeed, the implementation of specific protocols to educate mothers before discharge, such as providing informative graphic pamphlets and clear verbal instructions on when is necessary to present to a health care facility, could be an effective way to prevent clinical deterioration and delayed care, as demonstrated by Berhea and colleagues in Ethiopia [34]. We also found that severe malnutrition became a more frequent cause of hospitalization with increasing age, being the third cause of admission in preschoolers. This is an important finding, considering that it is estimated that malnutrition is the underlying cause of 45% of global deaths in children below 5 years of age [35, 36]. In the school-aged group, malaria was the leading cause of hospitalization followed by osteoarticular injuries, mainly due to falls and road accidents. This is consistent with previous studies in LMICs, reporting an increased frequency of injury in this age group [37–39]. Lastly, we also found an increased prevalence of haemato-oncological disorders in this age group compared to the others, mainly due to severe anemia.

Although data on presenting complaints were limited by the number of missing data, we found a similar distribution compared with previously published data from other low and middle-income countries (LMICs) [12, 40]. As expected, infection-related presenting complaints were the most frequent, with respiratory, fever, and gastrointestinal conditions being

the most common [41, 42]. After analyzing presentations complaints by age groups, it becomes evident how injury-related presentations increased with age, reaching almost half of the visits (48%) in school-age children. This is consistent with a previous Mozambican report [43]. However, within the neonatal and infant age groups, we found a high rate of injury-related presentations (16% and 14%, respectively), in particular, due to falls. These rates are in contrast with reports from high-income countries and other LMICs [39, 40]. Our rates are concerning, considering the fragility of children in this young age group and the overall high mortality and morbidity associated with injury [39]. Although further investigations are necessary to identify the reasons behind the abnormally high prevalence of injury in non-ambulant children found in this study, our results suggest that there is an urgent need to develop injury-prevention programs and campaigns to reduce injury rates in young Mozambican children, supporting the needs evidenced by De Sousa Petersburgo and coworkers [43].

Other useful interventions to improve the quality of care provided to the acutely and critically ill children include the implementation of a triage system, training of healthcare personnel and the establishment of an efficient emergency call and transport system. Several studies have shown how the introduction of Emergency Triage and Treatment (ETAT) guidelines could be an easy and cost-effective strategy to improve emergency and overall care [44–46]. In Malawi, the implementation of ETAT halved the pediatric inpatient mortality rate [47]. Training of health care personnel to the early identification of critical diseases is another important step in the improvement of PEC, as shown by several studies [10, 18, 48, 49]. Training on the early recognition and management of conditions that most often result in death in the local setting should be prioritized. Although challenging and expensive, the establishment of an efficient emergency call and transport by ambulance service would be critical to ensure that severely ill children from both the extra-urban and urban areas have access to the PED in a timely manner [31].

## Conclusions

Optimization of data registration is an important area on which to focus resources in order to obtain more accurate data to learn from and guide the development of pediatric emergency care in Mozambique, tailored to local needs. Our data provide insight into opportunities to reduce the high mortality in the pediatric emergency department and the high hospitalization rate, identifying the neonatal age group as the most vulnerable. Interventions such as maternal education, injury prevention, implementation of a triage system, training of health care personnel, and implementation of an emergency care transport system would be critical to improve the outcomes of acutely and critically ill children in Mozambique.

## Supporting information

**S1 Table. Determinants of hospitalization, mortality in the PED, and early death.** The valid cases (total of data available) have been reported with the number of hospitalized patients. Univariable analysis results are reported in the table.
(DOCX)

**S2 Table. Association between presenting complaints and hospitalization.** Absolute numbers and percentages of hospitalized patients have been reported. Univariable Analysis (OR, 95% Confidence Intervals (CI), and p-values, adjusted for gender, age, residency, and modality of presentation) are represented in the table.
(DOCX)

## Acknowledgments

Enrico Giordan M.D. for his support with the statistical analysis.

## Author Contributions

**Conceptualization:** Valentina Brugnolaro, Serena Calgaro, Liviana Da Dalt.

**Data curation:** Valentina Brugnolaro, Laura Nai Fovino, Serena Calgaro, Danila Azzolina.

**Formal analysis:** Valentina Brugnolaro, Dario Gregori, Danila Azzolina.

**Investigation:** Valentina Brugnolaro, Laura Nai Fovino.

**Methodology:** Valentina Brugnolaro, Serena Calgaro, Silvia Bressan, Liviana Da Dalt.

**Project administration:** Arlindo Rosario Muhelo.

**Resources:** Giovanni Putoto, Liviana Da Dalt.

**Software:** Valentina Brugnolaro.

**Supervision:** Valentina Brugnolaro, Dario Gregori, Silvia Bressan, Liviana Da Dalt.

**Validation:** Valentina Brugnolaro.

**Visualization:** Valentina Brugnolaro, Silvia Bressan.

**Writing – original draft:** Valentina Brugnolaro.

**Writing – review & editing:** Valentina Brugnolaro, Giovanni Putoto, Silvia Bressan, Liviana Da Dalt.

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
