## [Decision Letter · Decision Letter 0]

15 Jun 2020

PONE-D-20-12410

Pediatric emergency care in low income countries: characteristics and outcomes of presentations to a tertiary-care emergency department in Mozambique

PLOS ONE

Dear Dr. Brugnolaro,

Thank you for submitting your manuscript to PLOS ONE. After careful consideration, we feel that it has merit but does not fully meet PLOS ONE’s publication criteria as it currently stands. Therefore, we invite you to submit a revised version of the manuscript that addresses the points raised during the review process.

We look forward to receiving your revised manuscript.

Kind regards,

Itamar Ashkenazi

Academic Editor

PLOS ONE

Journal Requirements:

Reviewers' comments:

Reviewer's Responses to Questions

**Comments to the Author**

1. Is the manuscript technically sound, and do the data support the conclusions?

Reviewer #1: Yes

Reviewer #2: Yes

Reviewer #3: Partly

Reviewer #4: Partly

2. Has the statistical analysis been performed appropriately and rigorously? 

Reviewer #1: No

Reviewer #2: Yes

Reviewer #3: No

Reviewer #4: Yes

3. Have the authors made all data underlying the findings in their manuscript fully available?

Reviewer #1: Yes

Reviewer #2: No

Reviewer #3: Yes

Reviewer #4: Yes

4. Is the manuscript presented in an intelligible fashion and written in standard English?

Reviewer #1: Yes

Reviewer #2: Yes

Reviewer #3: Yes

Reviewer #4: Yes

5. Review Comments to the Author

Reviewer #1: - Materials and methods: The authors should specify if in registry of deaths are missing some data from the original 24844 and how many data are missed in the registry of hospitalization (we can assume that all the missed are discharged?)

- Statistical Analysis:

o The approval of the Ethics Committee is already written in Materials and Methods, where is more appropriated

o something in the data is incorrect: 396/24844 is 1,6% of mortality (1% is 247 <4h and 0,6% is 149 >4h) and hospitalization is 20,1% (4997/24844) and discharge 78,3% (19451/24844)

o It could be questionable that mortality rate is calculated in all population of 24844, because is not clear what’s happened to the children discharged (how long is the trip to home, what kind of assistance they have at home like general practitioner, nurse or other sanitary help, maybe some other information about the medical territorial system should be described in the introduction). Also, the description of the neonatology should be shortly written in the introduction (intensive unit is only pediatric? The hospital has a neonatal intensive care unit? How many beds if so?)

- Discussion:

o more emphasis about the missing records is necessary (only 14204 recorded at presentation means that more than 40% of the children arrived in ER could not have been properly evaluated and then deceased at home? Or only the children with more severe clinical presentation have been recorded?)

o Limitations in my opinion should be written before the discussion (mortality rate could be different for missing information i.e.)

Reviewer #2: The authors report a retrospective review of pediatric ED visits of >24,000 visits in Mozambique. They found that one third of ED visits were for injuries, ~20% of children were admitted from the ED, and 2% died in the ED. The strengths of this study are the large sample size, the clear writing, and the provision of the fundamental knowledge for future interventions for targeted improvements in care in Mozambique.

Despite the article’s strengths, there are several weaknesses that this reviewer thinks should be addressed. Some of these weaknesses include clearer interpretation of the data in the Tables, more of a discussion of how the large amount of missing data should affect their interpretation of the data, and the lack of data on interventions (or lack thereof) given in the ED. I have provided specific suggestions on ways to improve this important article below.

Title:

-I suggest changing “…in low income countries” to “in a low income country” as the study comes from a single low-income country.

Introduction:

-Consider avoiding discussing “critical care services” as they differ from emergency care.

-Final paragraph: sub-Saharan not “sub-Saharian”

-Minor point, but in the final paragraph the mortality rate (73 per 1,000) cited is a rate, not a probability.

-I suggest moving the final paragraph of the Introduction to the Methods as this is a description of the study setting and does little to emphasize the importance of the study. That being said, of course, the final two sentences in the final paragraph need to remain in the Introduction.

Methods:

-Did the authors gather data on average length of stay? I ask because some EDs in sub-Saharan Africa serve as hybrid EDs (in the Western sense) and ICUs as children who require critical care interventions stay in the ED because they lack stability for transfer to the actual ICU in the hospital.

-As the authors aim to understand emergency care and lay the groundwork for ways to improve care, was there any data on interventions, treatments, or resuscitations in the ED?

-A mention of the hospital mortality rate among children would be helpful to contextualize these findings that 2% of ED presentations resulted in death.

-Did the authors gather data on malnutrition, HIV, TB? These are important and well-documented associated factors with mortality among children.

Results:

-How does the fact that >1/3rd of all presenting complaints were missing influence the interpretation of the data?

-Table 2: I suggest listing the variables in descending order of frequency.

-Table 2: musculoskeletal is misspelled.

-Table 2: What are “sense organs” as a presenting complaint? I suggest including a footnote describing what those are and for all other disease categories in the Tables.

-I suggest including the exact number and exact percentage of patients who were admitted. “One-fifth of the presentations…” is not clear enough, especially since the following sentence cites a proportion of those who were admitted. It is helpful to have that actual number easily accessible.

-Table 3: It is unclear what the referent is for the multivariate analysis. For example, what exactly do the numbers on the same row as AGE mean? There should be a referent for each set of variables in the univariate and multivariate analysis.

-Table 3: aren’t the ORs under multivariate analysis adjusted ORs?

-In general, in the Results, the authors should be careful to clearly state the comparator for any comparison. For instance, “Children presenting for any kind of injury had a higher chance to be discharged except if they presented for burns.” Higher chance than what? Also, this should state higher odds.

-While almost 2/3rds of deaths occurred in the first 4 hours, what was the median time to death? This will help get at how long patients tend to stay in this ED.

-I suggest double checking the wording or stats cited in the sentence that contains, “significantly higher for children in the 0-28 days and in the 29 days - 1 year groups (OR: 3.58, 95% CI: 2.30 - 5.59, p-value < 0.001 and OR: 1.18, 95% CI 0.88 – 1.58, p-value = 0.283, respectively)” as the second OR is not significant.

-Same comment on Tables 6 and 7 regarding the need for a referent when reporting ORs.

-Table 7, no need to repeat column of Mortality in the PED as this was shown in a previous table.

-Why do the authors not take into account underlying medical problems in the multivariate analysis? Surely HIV, malnutrition, TB, cancer, etc. contribute to hospital admissions and death but these are not taken into account in the current analysis.

Discussion:

-Please define LMIC the first time it appears in the paper.

-Only 19% of reason for presentation among children who died? This should be discussed in the Limitations specifically as much of the Introduction and Discussion circles around mortality.

-I suggest adding to the Limitations that this was a single hospital and may not represent the rest of Mozambique or other LICs with different disease prevalences.

Reviewer #3: In the context of improving pediatric emergency care services among low-income countries, authors aimed at studying profiles and outcomes in a tertiary care PED in Mozambique.

As highlighted by the authors, even if results from the study could not be generalized, the research question is of interest because these results from the field of a PED would help in leading strategies to improve their PED.

Introduction: Well written, correctly documented and referenced. The research question is well exposed and argued. It could be added some additional description about Mozambique and his healthcare facilities in general (if possible) in order to better understand how is organized the patient pathway. Is there any difference between Beira and Sofala about healthcare facilities?

Materials and Methods: it is a retrospective study. Because there were 3 registries, data were abstracted by 2 investigators with a high risk of missing data. Because there are many subset of data with which statistical analysis were performed, it would be helpful if authors could add a flow chart on the way they obtained the different subsets.

Statistical analysis: results were exposed from many different subsets of data (Table 1, n = 24,844 / Table 2, n =14,204 / Table 3, n = 14,448 / Table 4, n = 14,204 / Table 5, n = 24844, Table 6, n = 14,448 / Table 7, n = 28,844).

Even if it is understandable to perform analysis using the largest size of available data according to type of analysis, wouldn’t it more readable and more understandable if results were coming from a unique data set that contain all the data?

Could the authors indicate how missing data were handle for statistical analysis?

Results section: According logistic regression analysis: odds ratio are exposed according to a reference category.

In many tables, could the authors explain why reference categories were not specify? As we could understand, authors exposed results using dichotomized variables (Yes versus No). For example, in Table 3, for the variable “Residency”, wouldn’t it be more readable if the item “urban” was set as a reference category? If yes, Authors should consider this question for all dichotomized variables (in table 3, 4, 5, 6, 7).

In table 4, all the presenting complaints are at higher risk to be hospitalized that is unusual. Did the authors test interaction between them? Authors should not be limited in presenting univariate analysis. In order to better understand, I would suggest the authors to complete those results by performing multivariate analysis including demographic variables.

In table 6 &7, it may be frustrating not to go further in the analysis. Replacing “Presenting complaints” (“Medical” versus “Injury”) by “Presenting Reason” (“Fever”, “Respiratory”, “neurological” etc…) would be helpful in order to point out a more detailed profiles of those patients at higher risk of mortality.

Discussion section: If possible, to understand the trend of the results, could authors explain why their results were compared with data from Nepal, Pakistan, Malawi, Ethiopia, etc…? Are the patient profiles comparable? What about comparing their healthcare facilities?

Reviewer #4: PONE-D-20-12410

Review

General thoughts:

- This is a retrospective chart review with a primary objective of determining systems-level interventions and design features for an effective pediatric emergency medicine system in a low-income (Mozambique) country. The manuscript is well written in general, with more description required primarily in the methods section.

- With such a broad perspective, it puzzles me as to why the authors only went back one year (April 2017-2018). Internal validity of data is difficult to determine as there is a possibility of bias from random yearly variations (as our current COVID-19 era has surely shown us), that cannot be excluded without more than one year of data for comparison.

- Would interesting to know of acuity of presentation, and proportion of children requiring resuscitation, given the (fortunately) lower prevalence of mortality in the PED.

- The authors have taken a systems framework to the implications of their results. It would seem appropriate to further analyze and present data from a QI/systems lens to truly capture the complexity of their data and how they interact. (i.e. SEIPS 2.0/3.0 system). This “next level” analysis would provide much greater context for their prevalence data.

- The novelty of this study lies in their data source (low income African PED). As written this manuscript is more akin to an annual report. What concrete suggestions do these data make leadership focus upon when presented with the data? What is the generalizability of these data to countries in similar situations?

Introduction:

- Would state Millennium Development Goals/Sustainable Development Goals are sourced from the UN, to give context and international standards to these statements.

Methods:

- A key threat to the external validity of your study is the quality of your data registries? Please provide more data on who is responsible for keeping registry data. Are ICD 9/10 codes used? How is data stored/accessed? Is there a possibility that data could be altered after the fact? Who cureates/owns the databases (ie government, hospital, etc).

- How were charts identified for review?

- Was there a research ethics board that approved this study? Please state in the manuscript earlier, instead of in the last line.

- Please provide more of a description of your data abstraction process. Were there standard forms? What data was abstracted? Who were the abstractors? Where they trained? Was their intermittent overlap of data abstraction to ensure interrelater reliability?

Results

- There is a substantial proportion (43%) where presenting complaint data was not available. Please provide rationale of missing data (this is an interesting finding in and of itself and should be embraced as a finding of the study).

- What is the rationale for the presenting reason categories? Are they ICD9/10 based?

- To help with the generalizability of the data, it would be useful to get information on how the PED is structured, size of division, training background.

- Table 4 – Injury – column 2 – would add % data in brackets.

Discussion

- Interesting points are contained. Somewhat disorganized and would take a systems analysis perspective lens. Consider SEIPS 2.0 or 3.0 models

6. PLOS authors have the option to publish the peer review history of their article (what does this mean?). If published, this will include your full peer review and any attached files.

Reviewer #1: No

Reviewer #2: No

Reviewer #3: Yes: Antoine TRAN

Reviewer #4: No

---

## [Author Response · Author response to Decision Letter 0]

18 Aug 2020

Dear Editor in Chief, Prof. Itamar Ashkenazi,

Thank you for your thorough and timely review of our manuscript "Pediatric emergency care in a low-income country: characteristics and outcomes of presentations to a tertiary-care emergency department in Mozambique". 

We have responded point by point to all the comments made by the reviewers in our rebuttal letter below. As per instructions we also uploaded a marked-up copy of our manuscript that highlights changes made to the original version as a separate file labeled 'Revised Manuscript with Track Changes'. We uploaded the unmarked version of our revised paper without tracked changes as a separate file labelled 'Manuscript'. However, following the reviewers’ suggestions, we have almost completely re-written the manuscript based on the input from co-authors, making it very challenging to submit a precise marked-up copy of our manuscript that highlights every changes made to the original version. For this reason, we ask the editors and reviewers to consider primarily the clean version of the revised manuscript. Moreover, we endeavored to be as specific as possible in our response to the reviewers below.

We are looking forward to hearing from you.

Sincerely yours.

Best regards,

Dr. Valentina Brugnolaro

(corresponding author)

Reviewer #1: 

- Materials and methods: The authors should specify if in registry of deaths are missing some data from the original 24844 and how many data are missed in the registry of hospitalization (we can assume that all the missed are discharged?)

Authors’ reply: We thank the Reviewer for the opportunity to clarify this point. The death and hospitalization registries of the pediatric emergency department (PED) only recorded data of patients who died in the PED and were hospitalized, respectively. This means that all the visits that were included in the presentations registry but were not recorded either in the hospitalization registry, or in the death registry were discharged to home. We hope we have been able to better clarify this point in the revised version of the Material and Methods section (subsection “Sources of data and data collection procedures”)

- Statistical Analysis:

o the approval of the Ethics Committee is already written in Materials and Methods, where is more appropriated

Authors’ reply: The Reviewer is right; this information was reported twice in the original manuscript. Following the Reviewer’s suggestion, we have removed the sentence on Ethics Committee approval from the Statistical Analysis section. 

- something in the data is incorrect: 396/24844 is 1,6% of mortality (1% is 247 <4h and 0,6% is 149 >4h) and hospitalization is 20,1% (4997/24844) and discharge 78,3% (19451/24844)

Authors’ reply: We thank the Reviewer for noticing this inconsistency in rounding decimal digits. We made the necessary adjustments, and further checked all the results throughout the revised manuscript. 

- It could be questionable that mortality rate is calculated in all population of 24844, because is not clear what's happened to the children discharged (how long is the trip to home, what kind of assistance they have at home like general practitioner, nurse or other sanitary help, maybe some other information about the medical territorial system should be described in the Introduction). Also, the description of the neonatology should be shortly written in the Introduction (intensive unit is only pediatric? The hospital has a neonatal intensive care unit? How many beds if so?)

Authors’ reply: We appreciate the Reviewer's comments on mortality rate. However, we aimed to describe the mortality in the PED, rather than the overall mortality of children accessing the PED after they have been discharged. This would be very valuable information to have to identify further areas of improvement in pediatric emergency care. Unfortunately, the current local system does not allow to retrieve data on mortality after discharge. Based on the Reviewer’s suggestion we have added more detailed information on the territorial healthcare system and on the different types of care facilities for children, such as the neonatal and intensive care settings (Introduction Page 3,, Paragraph 4 and Material and Methods Page 4, Paragraph Study Setting: Healthcare in Beira and the Sofala province)

-Discussion:

o more emphasis about the missing records is necessary (only 14204 recorded at presentation means that more than 40% of the children arrived in ER could not have been properly evaluated and then deceased at home? Or only the children with more severe clinical presentation have been recorded?)

Authors’ reply: We thank the Reviewer for the opportunity to better clarify this important point. Although death and hospitalization registries were filled by ED medical personnel, allowing for accurate reporting of data, the presentation registry was filled by administrative staff who were less aware of the importance of complete information recording. This led to demographic data to be reported in the registry for all visits, while presenting complaints were reported for only 14,204 visits. The rate of missing information on presenting complaints was higher for patients who died in the PED (presenting complaint was not reported in 81.3% of deaths), followed by visits resulting in discharge to home (48.2% of missing information) and those resulting in hospitalization (missing information for 18.8%). As reported in the response to a previous Reviewer’s comment above, we could not retrieve information on death after discharge. We agree this would be valuable information to have for a broader understanding of gaps in pediatric emergency care management. We have revised the limitations section to give more emphasis to this aspect (Limitation section Page 18, Paragraph 1, Line 281)

o Limitations, in my opinion, should be written before the discussion (mortality rate could be different for missing information i.e.)

Authors’ reply: We do understand the Reviewer’s point and following their suggestion we have moved the limitation section before the discussion in order to help the reader correctly interpret our findings before the start of the discussion. The missing information on presenting complaints was inherently related to the local data registration and collection system. As reported above, data on mortality are accurate and there is no missing information with respect to PED visits that resulted in death. The limitations section has been revised to better clarify these points.

Reviewer #2: 

The authors report a retrospective review of pediatric ED visits of >24,000 visits in Mozambique. They found that one third of ED visits were for injuries, ~20% of children were admitted from the ED, and 2% died in the ED. The strengths of this study are the large sample size, the clear writing, and the provision of the fundamental knowledge for future interventions for targeted improvements in care in Mozambique.

Despite the article's strengths, there are several weaknesses that this reviewer thinks should be addressed. Some of these weaknesses include clearer interpretation of the data in the Tables, more of a discussion of how the large amount of missing data should affect their interpretation of the data, and the lack of data on interventions (or lack thereof) given in the ED. I have provided specific suggestions on ways to improve this important article below.

Title:

-I suggest changing "…in low income countries" to "in a low income country" as the study comes from a single low-income country.

Authors’ reply: We agree with the comment made by the Reviewer and we have changed the title accordingly.

Introduction:

-Consider avoiding discussing "critical care services" as they differ from emergency care. 

Authors’ reply: Based on the Reviewer’s comment we replaced “critical care services” with “emergency care services”

-Final paragraph: sub-Saharan not "sub-Saharian"

Authors’ reply: Thank you for noticing this typo. We have made the suggested change

-Minor point, but in the final paragraph the mortality rate (73 per 1,000) cited is a rate, not a probability.

Authors’ reply: The Reviewer is right, we have amended the manuscript accordingly

-I suggest moving the final paragraph of the Introduction to the Methods as this is a description of the study setting and does little to emphasize the importance of the study. That being said, of course, the final two sentences in the final paragraph need to remain in the Introduction.

Authors’ reply: We thank the Reviewer for this suggestion. We have revised the Introduction according to the Reviewer's advice (paragraph moved to Material and Methods, Study setting, Page 4, Paragraph Study Setting, Line 79).

Methods:

-Did the authors gather data on average length of stay? I ask because some EDs in sub-Saharan Africa serve as hybrid EDs (in the Western sense) and ICUs as children who require critical care interventions stay in the ED because they lack stability for transfer to the actual ICU in the hospital.

Authors’ reply: We thank the Reviewer for their suggestion to expand on our analysis including the average length of stay. We have now included in the results section the data on the length of stay in the PED, which we could abstract from the hospitalization registry only. (Page 10, Paragraph Outcomes, Line 189). The PED at the HCB includes a room dedicated to Pediatric Intensive Care, where critical children are admitted once stabilized. This information is now included in the methods section of the revised manuscript. (Page 4, Paragraph Study Setting, Line 90). However, patients who were admitted in the Pediatric Intensive Care area were classified as hospitalized to PICU in the hospitalization registry. 

-As the authors aim to understand emergency care and lay the groundwork for ways to improve care, was there any data on interventions, treatments, or resuscitations in the ED? 

Authors’ reply: Unfortunately, we were unable to collect data on interventions, treatments, or resuscitations in the PED because this information was reported on paper forms, which were not filed systematically and were therefore unavailable to either clinical or research staff. We have added a paragraph in the Methods section to better clarify this point. (Page 4, Paragraph Sources of data and data collection procedures, Line 102).

-A mention of the hospital mortality rate among children would be helpful to contextualize these findings that 2% of ED presentations resulted in death. 

Authors’ reply: We thank the Reviewer for this suggestion. Based on the most recent available data from 2017, in-hospital overall pediatric mortality was 13%, while in the NICU mortality was 33%. We have now added this information in the Materials and Methods section, under the “Study Setting” paragraph. 

-Did the authors gather data on malnutrition, HIV, TB? These are important and well-documented associated factors with mortality among children.

Authors’ reply: We agree with the Reviewer that it will be very important to have data on malnutrition, HIV and TB. Unfortunately, these data were reported inconsistently in the hospitalization registry only and cannot be accurately summarized to reflect the actual frequency of these underlying conditions in patients presenting to the PED. We have added this as an additional limitation to the study in the Limitations section.

Results:

-How does the fact that >1/3rd of all presenting complaints were missing influence the interpretation of the data?

Authors’ reply: We thank the Reviewer for this comment. We have now better specified this point in the limitation section, which, based on the suggestion of Reviewer 1, has now been moved before the discussion in order to help the reader correctly interpret our findings before the start of the discussion. In addition, we re-run the multivariable analysis and performed a sensitivity analysis to better address how missing information affects our results.

-Table 2: I suggest listing the variables in descending order of frequency.

-Table 2: musculoskeletal is misspelled. 

-Table 2: What are "sense organs" as a presenting complaint? I suggest including a footnote describing what those are and for all other disease categories in the Tables. 

Authors’ reply: We thank the Reviewer for their suggestions to improve the quality of Table 2. We have made the suggested changes. We added the following footnote to the Table “Sense organs are defined as the body organs by which humans are able to see, smell, hear, taste and touch or feel. This category includes medical complaints to the eyes, ears, nose, throat and to the skin.”

-I suggest including the exact number and exact percentage of patients who were admitted. "One-fifth of the presentations…" is not clear enough, especially since the following sentence cites a proportion of those who were admitted. It is helpful to have that actual number easily accessible. 

Authors’ reply: We have amended the Results section according to the Reviewer’s suggestion (Page 10, Paragraph Outcomes, Line 188)

-Table 3: It is unclear what the referent is for the multivariate analysis. For example, what exactly do the numbers on the same row as AGE mean? There should be a referent for each set of variables in the univariate and multivariate analysis.

Authors’ reply: Based on the Reviewer’s comment we have involved experienced professional statisticians (Dr. Danila Azzolina and Prof. Dario Gregori) to re-run the multivariate analysis. All the Results section has been revised based on the new analysis. Given the substantial contribution of Dr. Danila Azzolina and Prof Dario Gregori in the analysis and revisions of the manuscript they have been included as authors in the manuscript.

-Table 3: aren't the ORs under multivariate analysis adjusted ORs? 

Authors’ reply: Please see our response to the comment above.

-In general, in the Results, the authors should be careful to clearly state the comparator for any comparison. For instance, "Children presenting for any kind of injury had a higher chance to be discharged except if they presented for burns." Higher chance than what? Also, this should state higher odds.

Authors’ reply: Based on the Reviewer’s comment we have revised the whole Results section to add clarity to the reporting of our findings, making sure the comparator is specified for any comparison.

-While almost 2/3rds of deaths occurred in the first 4 hours, what was the median time to death? This will help get at how long patients tend to stay in this ED. 

Authors’ reply: Unfortunately, the exact time of death was not reported in the death registry, as data were collected according to the predefined categories of early death < 4 hours and later death in the PED >4 hours. This was an intrinsic limitation related to the local data collection system.

-I suggest double checking the wording or stats cited in the sentence that contains, "significantly higher for children in the 0-28 days and in the 29 days - 1 year groups (OR: 3.58, 95% CI: 2.30 - 5.59, p-value < 0.001 and OR: 1.18, 95% CI 0.88 – 1.58, p-value = 0.283, respectively)" as the second OR is not significant.

Authors’ reply: We thank the Reviewer for noticing this imprecision in reporting of our results. As reported in the response to a previous Reviewer’s comment above we have thoroughly revised the whole Results section to ensure more clarity and consistency in data reporting. 

-Same comment on Tables 6 and 7 regarding the need for a referent when reporting ORs.

Authors’ reply: As reported above all the Results section, including all the analyses and Tables have been thoroughly revised. All the multivariate analyses now include a reference category for each of the independent predictor variable.

-Table 7, no need to repeat column of mortality in the PED as this was shown in a previous table.

Authors’ reply: Table 7 has been revised taking into account the Reviewer’s suggestion. 

-Why do the authors not take into account underlying medical problems in the multivariate analysis? Surely HIV, malnutrition, TB, cancer, etc. contribute to hospital admissions and death but these are not taken into account in the current analysis.

Authors’ reply: As reported in the response to a previous Reviewer’s comment above, unfortunately data on underlying medical problems were reported only in the hospitalization registry and inconsistently. This information cannot be accurately summarized to reflect the actual frequency of these underlying conditions in patients presenting to the PED and cannot be used in the multivariate analysis. We have added this as an additional limitation to the study in the Limitations section.

Discussion:

-Please define LMIC the first time it appears in the paper.

Authors’ reply: We thank the Reviewer for noticing the acronym was not defined in the text. We have amended the manuscript accordingly (discussion section).

-Only 19% of reason for presentation among children who died? This should be discussed in the Limitations specifically as much of the Introduction and Discussion circles around mortality. 

Authors’ reply: We agree with the Reviewer and we have expanded the limitation section to include this point. 

-I suggest adding to the Limitations that this was a single hospital and may not represent the rest of Mozambique or other LICs with different disease prevalences.

Authors’ reply: We agree with the Reviewer and we have expanded the limitation section to include this point. 

Reviewer #3: in the context of improving pediatric emergency care services among low-income countries, authors aimed at studying profiles and outcomes in a tertiary care PED in Mozambique.

As highlighted by the authors, even if results from the study could not be generalized, the research question is of interest because these results from the field of a PED would help in leading strategies to improve their PED.

Introduction: Well written, correctly documented and referenced. The research question is well exposed and argued. It could be added some additional description about Mozambique and his healthcare facilities in general (if possible) in order to better understand how is organized the patient pathway. Is there any difference between Beira and Sofala about healthcare facilities? 

Authors’ reply: We thank the reviewer for their positive comment. As suggested by the Reviewer we added a dedicate paragraph in the Material and Methods section “Study Setting: Healthcare in Beira and the Sofala district”

Materials and Methods: it is a retrospective study. Because there were 3 registries, data were abstracted by 2 investigators with a high risk of missing data. Because there are many subset of data with which statistical analysis were performed, it would be helpful if authors could add a flow chart on the way they obtained the different subsets.

Authors’ reply: We thank the reviewer for his advice. We added a figure to better describe the characteristics of data sources. (Figure 1)

Statistical analysis: results were exposed from many different subsets of data (Table 1, n = 24,844 / Table 2, n =14,204 / Table 3, n = 14,448 / Table 4, n = 14,204 / Table 5, n = 24844, Table 6, n = 14,448 / Table 7, n = 28,844). Even if it is understandable to perform analysis using the largest size of available data according to type of analysis, wouldn’t it more readable and more understandable if results were coming from a unique data set that contain all the data?

Authors’ reply: The Reviewer is right in that using different subsets of data may lead to confusion. However, presenting complaints were reported for only 14,204 visits. For analyses where data on presenting complaints were included, the subset of analysis was limited to this number of patients. We made sure there were only two subsets of data used for analysis throughout the manuscript, i.e. the total number of records 24,884 and those where information of presenting complaints was reported, namely 14,204 records. 

Could the authors indicate how missing data were handle for statistical analysis? 

Authors’ reply: We thank the reviewer for his comment. Based on the Reviewer’s comments we have involved experienced professional statisticians (Dr. Danila Azzolina and Prof Dario Gregori) to re-run all the analyses. All the Results section has been revised based on the new analysis. Given the substantial contribution of Dr. Danila Azzolina and Prof Dario Gregori in the analysis and revisions of the manuscript they have been included as authors in the manuscript. Missing data were not included in the statistical analysis. However, in the analysis on the determinants of hospitalization and death we ran a sensitivity analysis for missing data imputation on the variable “presenting complaint” classified as “medical” versus “injury”, and found similar results to the analysis run on the subset of 14,204 patients for which information on presenting complaint was available. Details of this analysis are now reported in the Statistical analysis section and results are included in the manuscript Tables.

Results section: According logistic regression analysis: odds ratio are exposed according to a reference category.

In many tables, could the authors explain why reference categories were not specify? As we could understand, authors exposed results using dichotomized variables (Yes versus No). 

Authors’ reply: Based on the Reviewer’s comment we have re-run the multivariate analyses. All the Results section has been thoroughly revised based on the new analysis. Please see our response to the Reviewer’s comment above.

For example, in Table 3, for the variable "Residency", wouldn't it be more readable if the item "urban" was set as a reference category? If yes, Authors should consider this question for all dichotomized variables (in table 3, 4, 5, 6, 7). 

Authors’ reply: We re-run all the multivariate analyses following the Reviewer’s suggestion.

In table 4, all the presenting complaints are at higher risk to be hospitalized that is unusual. Did the authors test interaction between them? Authors should not be limited in presenting univariate analysis. In order to better understand, I would suggest the authors to complete those results by performing multivariate analysis, including demographic variables.

Authors’ reply: Based on the Reviewer’s comments we have re-run the multivariate analyses. We have also analyzed in a separate multivariate analysis the influence of presenting complaints according to the medical and injuries sub-categories, as suggested (Table 5).

In table 6 &7, it may be frustrating not to go further in the analysis. Replacing "Presenting complaints" ("Medical" versus "Injury") by "Presenting Reason" ("Fever", "Respiratory", "neurological" etc…) would be helpful in order to point out a more detailed profiles of those patients at higher risk of mortality.

Authors’ reply: We agree with the Reviewer that it is frustrating not to be able to proceed further in the analysis of the association between presenting complaints and mortality due to the very high number of missing data on presenting complaints for visits that resulted in death (only 19% of these records had data on presenting complaints reported). Unfortunately we could not go further with the analysis in this sense. 

Discussion section: If possible, to understand the trend of the results, could authors explain why their results were compared with data from Nepal, Pakistan, Malawi, Ethiopia, etc…? Are the patient profiles comparable? What about comparing their healthcare facilities?

Authors’ reply: We understand the Reviewer’s point. However, as data on pediatric emergencies from low and middle income countries are very limited, we compared our findings with the available published data from other low-middle income countries. While comparison of healthcare facilities between different low-income countries goes beyond the purpose of our study, we focused our comparison on similarities on presenting complaints/outcomes between available pediatric datasets from the emergency department settings from different low-income countries. We have revised the discussion to better clarify the meaning of comparing our findings with data from other countries.

Reviewer #4: 

General thoughts:

- This is a retrospective chart review with a primary objective of determining systems-level interventions and design features for an effective pediatric emergency medicine system in a low-income (Mozambique) country. The manuscript is well written in general, with more description required primarily in the methods section.

- With such a broad perspective, it puzzles me as to why the authors only went back one year (April 2017-2018). Internal validity of data is difficult to determine as there is a possibility of bias from random yearly variations (as our current COVID-19 era has surely shown us), that cannot be excluded without more than one year of data for comparison.

Authors’ reply: We agree with the Reviewer’s comment. Unfortunately, based on the local clinical registry filing system, completed paper registries were temporarily filed for some months, and then periodically burnt. For this reason we did not have access to registries including data before April 2017. We have now specified this in the limitation section.

- Would interesting to know of acuity of presentation, and proportion of children requiring resuscitation, given the (fortunately) lower prevalence of mortality in the PED.

Authors’ reply: Unfortunately, we were unable to collect data on interventions, treatments, or resuscitations in the PED because this information was reported on paper forms, which were not filed systematically and were therefore unavailable to either clinical or research staff. We have added a paragraph in the Methods section to better clarify this point. (Page 4 Paragraph Sources of data and data collection procedures, Line 102).

- The authors have taken a systems framework to the implications of their results. It would seem appropriate to further analyze and present data from a QI/systems lens to truly capture the complexity of their data and how they interact. (i.e. SEIPS 2.0/3.0 system). This "next level" analysis would provide much greater context for their prevalence data.

Authors’ reply: We thank the Reviewer for this suggestion. Having complete and accurate data to be presented from a QI/system lens perspective would be ideal. However, our data specifically focus on the PED and, at this initial stage, they are able to offer insight into possible interventions at different stages of care within an integrated system of care that may be effective in reducing mortality in the PED and hospitalization. We hope that a more accurate and structured data monitoring system, linking the different stages and areas of care will be able to provide the appropriate data for a more complex and informative “next level” analysis such as using the SEIPS framework.

- The novelty of this study lies in their data source (low income African PED). As written this manuscript is more akin to an annual report. What concrete suggestions do these data make leadership focus upon when presented with the data? What is the generalizability of these data to countries in similar situations?

Authors’ reply: Based on the Reviewer’s comment we have thoroughly revised the discussion section and partially the conclusions in order to better report concrete suggestions for improvement in care. We have also expanded the limitations section to address the issue of generalizability.

Introduction:

- Would state Millennium Development Goals/Sustainable Development Goals are sourced from the UN, to give context and international standards to these statements.

Authors’ reply: We have amended the Introduction according to the Reviewer’s suggestion.

Methods:

- A key threat to the external validity of your study is the quality of your data registries? Please provide more data on who is responsible for keeping registry data. Are ICD 9/10 codes used? How is data stored/accessed? Is there a possibility that data could be altered after the fact? Who cureates/owns the databases (ie government, hospital, etc).

Authors’ reply: We thank the Reviewer for the opportunity to better clarify this point. Death and hospitalization registries were filled by ED medical personnel, while the presentation registry was filled by hospital administrative personnel. Data were collected according to local pre-defined categories for presenting complaints and hospitalization diagnosis. ICD 9/10 codes were not used. Based on the local clinical registry filing system, completed paper registries were temporarily filed in the PED for some months, and then periodically burnt. The registries were owned by the hospital. All this information has been reported in the revised version of the manuscript under the Material and Methods and Limitations sections. 

- How were charts identified for review?

Authors’ reply: Data were not abstracted by medical patients’ charts but only from the registries. Clinical notes and patients’ charts were not systematically filed for PED visits and therefore were not available for review. We hope the additional information on data collection we included in the revised version of the paper helps better clarify this point.

- Was there a research ethics board that approved this study? Please state in the manuscript earlier, instead of in the last line.

Authors’ reply: The study was indeed approved by the hospital ethics board. This piece of information is reported at the start of the Materials and Methods section.

- Please provide more of a description of your data abstraction process. Were there standard forms? What data was abstracted? Who were the abstractors? Where they trained? Was their intermittent overlap of data abstraction to ensure interrater reliability?

Authors’ reply: Data were abstracted by the paper registries in an electronic standard data collection system according to a predefined coding. All data systematically reported in the registries were abstracted. Abstractors were trained locally based on the initial review of 200 registry records each and a two-month data abstraction overlap between the abstractors helped in ensuring consistency in data abstraction and coding. However, formal interrater reliability was not calculated. This information has been better detailed in the Material and Methods section of the revised manuscript.

Results 

- There is a substantial proportion (43%) where presenting complaint data was not available. Please provide rationale of missing data (this is an interesting finding in and of itself and should be embraced as a finding of the study). 

Authors’ reply: We agree with the Reviewer that this point deserves further attention. Based on the Reviewer’s suggestion we have now discussed this as a finding of the study, early on in the discussion section. We have also expanded the Methods and Limitations sections to provide the rationale of the high proportion of missing data and their implications for the study and for the development of a pediatric emergency care system, which should be tailored to the local setting and needs, as reflected by local data, staff experience and patients’ perspectives.

- What is the rationale for the presenting reason categories? Are they ICD9/10 based? 

Authors’ reply: Data in the registries were collected according to local predefined categories. This information has been included in the revised version of the Material and Methods section. Unfortunately the HBC did not have the facilities and resources (i.e., trained staff, IT infrastructures) to collect data according to the ICD 9/10 codes.

- To help with the generalizability of the data, it would be useful to get information on how the PED is structured, size of division, training background. 

Authors’ reply: We thank the Reviewer for this suggestion. We have revised the Material and Methods section to include this information.

- Table 4 – Injury – column 2 – would add % data in brackets. 

Authors’ reply: We have amended Table 4 according to the Reviewer’s suggestion.

Discussion

- Interesting points are contained. Somewhat disorganized and would take a systems analysis perspective lens. Consider SEIPS 2.0 or 3.0 models

Authors’ reply: We thank the Reviewer for their suggestion. We have revised the discussion to improve its flow. Please see, our response to the comment above about SEIPS 2.0 or 3.0 models.

---

## [Decision Letter · Decision Letter 1]

14 Sep 2020

PONE-D-20-12410R1

Pediatric emergency care in a low income country: characteristics and outcomes of presentations to a tertiary-care emergency department in Mozambique

PLOS ONE

Dear Dr. Brugnolaro,

Thank you for submitting your manuscript to PLOS ONE. Three reviewers submitted their reviews and all three favored publishing your manuscript.  There still some minor comments the reviewers thought are appropriate and if dealt with will improve your manuscript further.  Therefore, we invite you to submit a revised version of the manuscript that addresses these minor points raised during the review process.

We look forward to receiving your revised manuscript.

Kind regards,

Itamar Ashkenazi

Academic Editor

PLOS ONE

Reviewers' comments:

Reviewer's Responses to Questions

**Comments to the Author**

1. If the authors have adequately addressed your comments raised in a previous round of review and you feel that this manuscript is now acceptable for publication, you may indicate that here to bypass the “Comments to the Author” section, enter your conflict of interest statement in the “Confidential to Editor” section, and submit your "Accept" recommendation.

Reviewer #2: All comments have been addressed

Reviewer #3: All comments have been addressed

Reviewer #4: All comments have been addressed

2. Is the manuscript technically sound, and do the data support the conclusions?

Reviewer #2: Yes

Reviewer #3: Yes

Reviewer #4: Yes

3. Has the statistical analysis been performed appropriately and rigorously? 

Reviewer #2: Yes

Reviewer #3: Yes

Reviewer #4: Yes

4. Have the authors made all data underlying the findings in their manuscript fully available?

Reviewer #2: Yes

Reviewer #3: Yes

Reviewer #4: Yes

5. Is the manuscript presented in an intelligible fashion and written in standard English?

Reviewer #2: Yes

Reviewer #3: Yes

Reviewer #4: Yes

6. Review Comments to the Author

Reviewer #2: The authors have been very responsive to reviewers’ comments. They have performed additional analyses and have been very thorough in assessing the limitations of their study. I applaud the authors for their responsiveness. I think the article is much strengthened at this point and I feel this is a good beginning to further elucidate what type of patients and outcomes occur in a single PED in Mozambique. My comments are extremely minor at this point.

-If the authors had a “two-month data abstraction overlap between the abstractors helped in ensuring consistency in data abstraction and coding”, should there be a kappa calculated to determine the inter-rater agreement?

-What was the p-value cut off the authors used to determine “significant” in the univariate model to then include in the multivariable model? Could be added around line 142.

-Line 166: “already been” could be reworded. Consider “42% had been assessed by a health care provider…”

-Consider presenting the variables in Tables 4 and 6 on different rows to make it clearer which variable is being reported on some lines. The authors appear to use bold text to do this, but I think splitting variables like “Health care provider referral” and “Self presentation” as separate rows. This will help with the multivariate reporting to as it will emphasize what the referent is.

Reviewer #3: The author has well answered point by point to all the comments. Since, the manuscript has been substantially improved and I don't have any additional comments to the author.

Reviewer #4: General thoughts:

- This is a retrospective chart review with a primary objective of determining systems-level interventions and design features for an effective pediatric emergency medicine system in a low-income (Mozambique) country. The authors responses were thorough and appreciated.

Abstract:

- A major finding in this paper is the lack of a “paper trial” or system for data records (i.e 43% of presenting complaint data is missing, the lack of multiple years of data to review). This finding shouldn’t simply be relegated to a limitation of the study. It should be a finding that is embraced as a major barrier to improving the care of children presenting for PED care in this country. I would love to see this finding reported on the abstract.

Materials and Methods:

- This revised section flows much better. The inclusion of information about the ward set-up and layout is of the PED and the Pediatric Healthcare system is much appreciated, and flows well.

- There seem to be two captions for Figure 1 (place clarify where the location of this figure is meant to be placed. The title for Figure 1 seems like it could be more like a result was reported. I would include the keyword: registries (i.e. something to the effect of Figure 1: Study register characteristics.

- We are unable to see Figure 1, and hence cannot comment on it.

- Please include in the manuscript your rationale for why ICD9/10 codes are not used.

- The new sensitivity analysis is a nice touch.

o Typo in point 3. Should read “outcomes”, not “otcomes”

Results:

- Please check spacing of the first row “mortality” in Table 3. Please make sure all the chart lines up (in admission section)

- Data is presented well an in an organized fashion. Tables are excellent.

- Limitations is presented in the Results section (page 19). Confusing to same it the same name as a section usually found in the discussion. Would rename this subtitle according to the primary finding regarding limitations of data sources (which is not a limitation of the methods and study design per se, but a separate stand alone finding).

Discussion:

- Rewritten discussion and focus on recommendations from data is much better understood. The relation to WHO and UNICEF guidelines is a good direction to take.

7. PLOS authors have the option to publish the peer review history of their article (what does this mean?). If published, this will include your full peer review and any attached files.

Reviewer #2: No

Reviewer #3: No

Reviewer #4: No

---

## [Author Response · Author response to Decision Letter 1]

2 Oct 2020

Dear Editor in Chief, Prof Itamar Ashkenazi,

Thank you for your thorough and timely review of our revised manuscript "Pediatric emergency care in a low-income country: characteristics and outcomes of presentations to a tertiary-care emergency department in Mozambique". 

We have responded point by point to all the comments made by the reviewers in our rebuttal letter below. As per instructions, we also uploaded a marked-up copy as a separate file labeled 'Revised Manuscript with Track Changes'. We uploaded the clean version of our revised paper without tracked changes as a separate file labelled 'Manuscript'.

We are looking forward to hearing from you.

Sincerely yours.

Best regards,

Dr. Valentina Brugnolaro

(corresponding author)

Reviewers' comments:

Review Comments to the Author

Reviewer #2: The authors have been very responsive to reviewers’ comments. They have performed additional analyses and have been very thorough in assessing the limitations of their study. I applaud the authors for their responsiveness. I think the article is much strengthened at this point and I feel this is a good beginning to further elucidate what type of patients and outcomes occur in a single PED in Mozambique. My comments are extremely minor at this point.

Author’s reply: We thank the reviewer for the supportive comment.

-If the authors had a “two-month data abstraction overlap between the abstractors helped in ensuring consistency in data abstraction and coding”, should there be a kappa calculated to determine the inter-rater agreement?

Author’s reply: We thank the reviewer for this comment. The two-month data abstraction overlap was used for training purposes of the second data abstractor, and for clarification of possible doubts. There was no double entry of all data, so unfortunately, we cannot determine the inter-rater agreement. To clarify this point, we added the following wording to the sentence "...consistency in data abstraction and coding, by training of the second data abstractor. No formal double entry of data by the two abstractors occurred during this time.”

-What was the p-value cut off the authors used to determine “significant” in the univariate model to then include in the multivariable model? Could be added around line 142.

Author’s reply: The p-value cut-off we used to include significant variables at the univariate model into the multivariable model was 0.05. We have added this information to the manuscript, as suggested by the Reviewer.

-Line 166: “already been” could be reworded. Consider “42% had been assessed by a health care provider…”

Author’s reply: We have revised the sentence accordingly.

-Consider presenting the variables in Tables 4 and 6 on different rows to make it clearer which variable is being reported on some lines. The authors appear to use bold text to do this, but I think splitting variables like “Health care provider referral” and “Self-presentation” as separate rows. This will help with the multivariate reporting to as it will emphasize what the referent is.

Author’s reply: We have revised tables 4 and 6 according to the reviewer suggestions.

Reviewer #3: The author has well answered point by point to all the comments. Since, the manuscript has been substantially improved and I don't have any additional comments to the author.

Author’s reply: We thank the reviewer for the supportive comment.

Reviewer #4: General thoughts:

- This is a retrospective chart review with a primary objective of determining systems-level interventions and design features for an effective pediatric emergency medicine system in a low-income (Mozambique) country. The authors responses were thorough and appreciated.

Author’s reply: We thank the reviewer for the supportive comment.

Abstract:

- A major finding in this paper is the lack of a “paper trial” or system for data records (i.e 43% of presenting complaint data is missing, the lack of multiple years of data to review). This finding shouldn’t simply be relegated to a limitation of the study. It should be a finding that is embraced as a major barrier to improving the care of children presenting for PED care in this country. I would love to see this finding reported on the abstract.

Author’s reply: As suggested, we have now included this finding in the Results section of the abstract and commented on it in the conclusion section of the abstract. 

Materials and Methods:

- This revised section flows much better. The inclusion of information about the ward set-up and layout is of the PED and the Pediatric Healthcare system is much appreciated and flows well.

Author’s reply: We thank the reviewer for the supportive comment.

- There seem to be two captions for Figure 1 (place clarify where the location of this figure is meant to be placed. The title for Figure 1 seems like it could be more like a result was reported. I would include the keyword: registries (i.e. something to the effect of Figure 1: Study register characteristics.

Author’s reply: The Reviewer is right, in the marked-up copy of the manuscript the caption appears twice. We realized that the paragraph referring to Figure 1 is duplicated in this version of the manuscript, while the clean version is correct and the Figure caption appears only once. As suggested, we have reworded Figure 1 caption as follows: "Characteristics of hospital registries from which study data were collected".

- We are unable to see Figure 1, and hence cannot comment on it.

Author’s reply: We are sorry that the Reviewer was unable to see figure 1. We followed the instructions for the upload of figures and could see it in the pdf of the manuscript for download. We hope there are no technical issues preventing the inclusion on the figure in the revised version of the manuscript. 

- Please include in the manuscript your rationale for why ICD9/10 codes are not used.

Author’s reply: As recommended by the Reviewer we have included the following sentence in the revised version of the manuscript, at the end of the section on "Sources of data and data collection procedures": "Unfortunately the HBC did not have the facilities and resources (i.e., trained staff, information technology infrastructure) to code diagnosis according to the ICD 9/10 codes. Data on diagnosis were reported as per local documentation practices". 

- The new sensitivity analysis is a nice touch.

Author’s reply: We thank the reviewer for the positive comment.

o Typo in point 3. Should read “outcomes”, not “otcomes”

Author’s reply: Thank you for noticing it, we have corrected the typo accordingly (line 154).

Results:

- Please check spacing of the first row “mortality” in Table 3. Please make sure all the chart lines up (in admission section)

Author’s reply: We thank the reviewer for noticing this, we have corrected the spacing accordingly (Table 3)

- Data is presented well an in an organized fashion. Tables are excellent.

Author’s reply: We thank the reviewer for the positive comment.

- Limitations is presented in the Results section (page 19). Confusing to same it the same name as a section usually found in the discussion. Would rename this subtitle according to the primary finding regarding limitations of data sources (which is not a limitation of the methods and study design per se, but a separate stand-alone finding).

Author’s reply: We thank the Reviewer for the suggestion. The limitation section was adjusted and reported just after the Results and before the Discussion following a previous recommendation from another Reviewer. We are aware some Journals prefer for the Limitations section to appear before the discussion rather than at the end of the discussion. We are happy to follow the Editors' recommendations as to where the Limitations section should be placed in the manuscript.

Discussion:

- Rewritten discussion and focus on recommendations from data is much better understood. The relation to WHO and UNICEF guidelines is a good direction to take.

Author’s reply: We thank the reviewer for the supportive comment.

---

## [Editor Report · Decision Letter 2]

12 Oct 2020

Pediatric emergency care in a low income country: characteristics and outcomes of presentations to a tertiary-care emergency department in Mozambique

PONE-D-20-12410R2

Dear Dr. Brugnolaro,

We’re pleased to inform you that your manuscript has been judged scientifically suitable for publication and will be formally accepted for publication once it meets all outstanding technical requirements.

Kind regards,

Itamar Ashkenazi

Academic Editor

PLOS ONE
---

## [Editor Report · Acceptance letter]

22 Oct 2020

PONE-D-20-12410R2 

Pediatric emergency care in a low-income country: characteristics and outcomes of presentations to a tertiary-care emergency department in Mozambique 

Dear Dr. Brugnolaro:

I'm pleased to inform you that your manuscript has been deemed suitable for publication in PLOS ONE. Congratulations! Your manuscript is now with our production department. 

Kind regards, 

on behalf of

Dr. Itamar Ashkenazi 

Academic Editor

PLOS ONE